# Risk for Recurrence in Long-Term Follow-Up of Children after Liver Transplantation for Hepatoblastoma or Hepatocellular Carcinoma

**DOI:** 10.3390/children11020193

**Published:** 2024-02-03

**Authors:** Marek Stefanowicz, Piotr Kaliciński, Hor Ismail, Adam Kowalski, Dorota Broniszczak, Marek Szymczak, Katarzyna Pankowska-Woźniak, Anna Roszkiewicz, Ewa Święszkowska, Diana Kamińska, Sylwia Szymańska, Grzegorz Kowalewski

**Affiliations:** 1Department of Pediatric Surgery and Organ Transplantation, The Children’s Memorial Health Institute, 04-730 Warsaw, Poland; m.stefanowicz@ipczd.pl (M.S.); h.ismail@ipczd.pl (H.I.); a.kowalski@ipczd.pl (A.K.); d.broniszczak@ipczd.pl (D.B.); m.szymczak@ipczd.pl (M.S.); k.pankowska-wozniak@ipczd.pl (K.P.-W.); a.roszkiewicz@ipczd.pl (A.R.); g.kowalewski@ipczd.pl (G.K.); 2Department of Oncology, The Children’s Memorial Health Institute, 04-730 Warsaw, Poland; e.swieszkowska@ipczd.pl; 3Department of Gastroenterology, Hepatology, Nutritional Disorders and Pediatrics, The Children’s Memorial Health Institute, 04-730 Warsaw, Poland; d.kaminska@ipczd.pl; 4Department of Pathomorphology, The Children’s Memorial Health Institute, 04-730 Warsaw, Poland; s.szymanska@ipczd.pl

**Keywords:** hepatoblastoma, hepatocellular carcinoma, liver transplantation

## Abstract

The aim of this study was to assess the long-term results of liver transplantation (LT) in pediatric patients with unresectable hepatoblastoma (HB) or hepatocellular carcinoma (HCC) with special reference to the risk of tumor recurrence. We retrospectively analyzed data from 46 HB and 26 HCC patients who underwent LT between 1990 and 2022. In HCC patients, we compared outcomes depending on donor type. We evaluated the impact of a number of risk factors on recurrence-free survival after LT. Estimated patient survival after 5, 10, and 15 years was 82%, 73%, and 73% in the HB group and 79%, 75%, and 75% in the HCC group, respectively (*p* = 0.76). In the HCC group, living donor LT (LDLT) and deceased donor LT (DDLT) provided similar patient survival (*p* = 0.09). Estimated recurrence-free survival in patients who had three or fewer risk factors was significantly better than in patients with more than three risk factors (*p* = 0.0001). Adequate patient selection is necessary when considering LT for primary liver tumors in children. The presence of more than three risk factors is associated with a very high risk of recurrence and indicates poor prognosis, whereas extrahepatic disease may be considered a contraindication for transplantation.

## 1. Introduction

In the pediatric population, hepatoblastoma (HB) is the most common primary malignant liver tumor, followed by hepatocellular carcinoma (HCC) [1]. Treatment of these tumors remains a significant clinical challenge. A fifth of all patients with HB require LT, whereas resection at diagnosis is not possible in most pediatric patients with HCC [2,3]. Patient survival without recurrence depends on complete surgical resection, especially in HCC, which is often resistant to chemotherapy. In patients with unresectable tumors, total hepatectomy during liver transplantation (LT) allows for complete surgical resection and improved survival of children with HB and HCC [4]. LT improves the survival of patients with HCC and underlying liver diseases [5], while living donor LT (LDLT) has been shown to offer better survival outcomes than deceased donor LT [1]. 

Our study assessed the long-term results of LT in pediatric patients with unresectable HB or HCC, focusing especially on the causes and patterns of tumor recurrence. 

## 2. Materials and Methods

We retrospectively analyzed data of pediatric patients diagnosed with HB or HCC who underwent LT at our institution between 3 March 1990 and 1 December 2022. The cohort was continuously evaluated, with the latest follow-up on 1 December 2023. 

Patients were categorized by diagnosis into groups, HB and HCC. In the HCC group, we also compared patient outcomes depending on donor type (deceased or living). In the HB group, the number of patients after deceased donor LT (DDLT) was insufficient to perform a similar analysis. 

Patient data were collected, including age at diagnosis and LT, gender, serum alpha-fetoprotein (AFP) level at diagnosis and before LT, underlying diseases, and hepatic resections before LT. Rescue LT was defined as transplantation in patients after surgical tumor resection and recurrence in the remaining liver. For oncologic evaluations, collected data included pretreatment extent of disease (PRETEXT), tumor multifocality or rupture, metastasis at diagnosis, presence of metastases after partial liver resection, and number of patients who received chemotherapy before and after LT. Collected data also included the presence of extrahepatic extension, lymph node metastases, and tumor pathology. The extrahepatic extension was defined as the tumor crossing boundaries/tissue planes. Angioinvasion was defined as the presence of tumor cells in the vessels. We analyzed surgical variables such as donor type, graft type, and biliary anastomosis. Outcome included event-free survival (EFS), defined as survival without tumor recurrence or graft loss (censored for patient death) and survival. We analyzed risk factors associated with tumor recurrence, time and site of relapse, treatment, and outcome to evaluate disease relapse. 

We evaluated the impact of a number of different risk factors on recurrence-free survival after LT in patients with HB and HCC. In this analysis, we used the risk factors for potential poor outcomes often discussed in the literature [1,6,7,8,9]. Our analysis included rescue LT, PRETEXT IV, metastases at diagnosis, tumor multifocality, tumor rupture, extrahepatic disease (defined as the presence of metastases after partial liver resection, presence of extrahepatic extension or lymph node metastases found during LT, or a positive margin after total hepatectomy), presence of angioinvasion, and unfavorable histology of tumors (macrotrabecular and small-cell undifferentiated pattern for HB, and fibrolamellar for HCC).

LT for the treatment of HB or HCC was performed in patients with unresectable tumors (PRETEXT IV or major vascular involvement) after neoadjuvant chemotherapy or in patients with chronic liver disease leading to cirrhosis and HB or HCC. The preoperative chemotherapy regimen included cisplatin and doxorubicin. Before LT, complete remission of metastatic disease was obtained. Qualification for LT was confirmed by a multidisciplinary team. Rescue LT was considered in selected patients.

The surgical technique in LDLT has been previously described [10]. For DDLT, we used the standard technique that Starlz et al. described with resection of the retrohepatic vena cava and two end-to-end anastomoses [11]. During LT, before starting hepatectomy, the abdominal cavity was evaluated to determine the extent of the disease, exclude the possibility of partial resection, and assess the feasibility of total liver resection. Indication for caval replacement during LDLT was assessed based on preoperative radiologic evaluation—ultrasound (US) examination and computer tomography (CT)—and confirmed during hepatectomy. For reconstruction, frozen grafts from the iliac vein were used.

Immunosuppression after LT due to HB or HCC consisted of tacrolimus and steroids. Children older than two years with ABO-incompatible LT received induction therapy with an Il-2 receptor antagonist. After LT, patients with HB and HCC received two or three courses of cisplatin and doxorubicin as standard treatment. 

AFP measurement and US examination were performed during follow-up at every check-up and when clinically indicated. CT of the abdomen and thorax was performed after the last course of adjuvant chemotherapy and 1 year after LT or if clinically indicated. 

The collected data were analyzed using Statistica 13.3 (TIBCO software). Continuous data were presented as median values and the interquartile range (IQR). We created survival curves using the Kaplan–Meier method to evaluate clinical outcomes and used a log-rank test to compare them. The chi-square test of independence was used for categorical data analysis, and the Mann–Whitney U test was used for continuous variables. Univariate and multivariable analyses were performed using Cox’s proportional hazard model. 

Prior to the study, we received the approval of the institutional Ethics Committee (approval number: 44/KBE/2021).

## 3. Results

Between 3 March 1990 and 1 December 2022, 819 patients younger than 18 underwent 892 LT, including 490 (50.4%) from living donors. Among these, 46 patients (5.6%) underwent LT for HB and 26 (3.2%) for HCC. 

### 3.1. HB Group (Table 1)

#### 3.1.1. Pretransplant Data

The median age at diagnosis was 1.5 years (IQR: 0.7 to 2.4 years). Twenty recipients (43.5%) in the HB group were female. Four patients (8.75%) had an underlying disease. One patient had Beckwith–Widemann syndrome, one had Abernethy syndrome, one had hepatitis C virus infection, and one had hepatitis B virus infection.

**Table 1 children-11-00193-t001:** Demographics and characteristics of the 46 patients with HB after LT.

Age at diagnosis	
Median (IQR)	1.5 years (0.7–2.4 years)
Age ≥ 8	4 (8.7%)
Gender	20 female (43.5%)
Underlying disease	4 (8.7%)
Beckwith–Widemann syndrome	1
Abernethy syndrome	1
Hepatitis C virus infection	1
Hepatitis B virus infection	1
Rescue LT	3 (6.5%)
AFP at diagnosis	
Median (IQR)	282,592 ng/mL (86,989–557,218 ng/mL)
<100 ng/mL	1
AFP pretransplant	
Median (IQR)	196 ng/mL (86,989–557,218 ng/mL)
Decline of AFP level before LT/at diagnosis > 95% ^1^	34/43 (79.1%)
Tumor characteristics	
PRETEXT I	1 (2.2%)
PRETEXT II	2 (4.3%)
PRETEXT III	22 (47.8%)
PRETEXT IV	21 (45.7%)
Multifocal tumor	20 (43.5%)
Metastases at diagnosis	10 (21.7%)
Lung	9
Mediastinal lymph nodule	1
Tumor rupture before LT	1 (2.2%)
Chemotherapy before LT	45 (97.8%)
Time from diagnosis to LT Median (IQR)	4.8 months (3.6–6 months)
Waitlist time Median (IQR)	19 days (7–42 days)
Age at LT	
Median (IQR)	1.9 years (1–2.8 years)
Graft type	
Living donor	41 (89.1%)
Left lateral segments	39
Left lobe	2
Deceased donor	5 (10.9%)
Whole liver	4
Reduced graft (left lobe)	1
Caval replacement (LDLT)	4/41 (9.8%)
Duct to duct biliary anastomosis	12 (26.1%)
ABO-incompatible LT	11 (23.9%)
Histology	
Epithelial–mesenchymal type	12
Epithelial-type fetal pattern	20
Epithelial-type embryonal type	4
Epithelial-type fetal–embryonal pattern	3
Epithelial-type macrotrabecular pattern	2
Epithelial-type small-cell undifferentiated pattern	1
Mesenchymal type	1
Not otherwise specified (NOS)	3
R0 margin	44 (95.7%)
Extrahepatic extension	2 (4.3%)
Angioinvasion	16 (34.8%)
Cirrhosis	1 (2.2%)
Chemotherapy after LT	42 (91.3%)

^1^ Patients after rescue LT were excluded from analysis.

There were three rescue LTs in patients after previous tumor resection and recurrence in the remaining liver. These patients had undergone right extended hemihepatectomy 24, 10, and 5 months before LT (Table 2). In the third patient, thermoablation of focal lesions in the remnant liver was performed twice, two and one months before LT. One patient needed urgent liver transplantation due to respiratory failure caused by a tumor mass. 

The median AFP at diagnosis was 282,592 ng/mL (IQR: 86,989 to 557,218 ng/mL). One patient had AFP below 100 ng/mL at diagnosis. The median AFP before LT was 196 ng/mL (IQR: 50 to 2230 ng/mL). 

Twenty-one patients (45.7%) with HB were classified as PRETEXT IV, 22 (47.8%) were PRETEXT III, two (4.3%) were PRETEXT II, and one (2.2%) was PRETEXT I. The patient with PRETEXT I and Abernethy syndrome had undergone embolization of a portosystemic shunt 3 years before LT and later developed cirrhosis and HB. Two patients with PRETEXT II were qualified for LT due to tumor recurrence after previous hepatic resections in another center. Multifocal tumor was detected in 20 children (43.5%). Metastases at diagnosis were found in 10 patients (21.7%): nine in the lungs and in one in a mediastinal lymph node.

Forty-five patients (47.8%) received preoperative chemotherapy. In 34 out of 43 children who received primary LT, a decline of AFP > 95% was observed after pretransplant chemotherapy. In all patients, metastases disappeared after chemotherapy; metastasectomy before LT was not necessary in any patient.

#### 3.1.2. Transplant Data and Tumor Pathology

The median time from diagnosis to LT was 1.1 years (IQR: 0.7 to 2.7 years), and the median waitlist time was 19 days (IQR: 7 to 24 days). The median age at LT was 1.9 years (IQR 1 to 2.8 years). 

Forty-one patients (89.1%) with HB received a living donor transplant. Two patients received a left lobe, and 39 children received a left lateral segment. Five patients (10.9%) underwent LT from a deceased donor. Four patients received a whole liver, and one, a graft reduced to the left lobe. In four children who underwent LDLT, vena cava resection was performed due to tumor proximity. ABO-incompatible LT was performed in 11 patients (23.9%). 

Extrahepatic extension was found in two patients during hepatectomy. In the first patient, who underwent rescue LT, infiltration of the spleen was seen and en bloc resection of the liver and spleen was performed to provide an R0 margin. In the second patient, who needed urgent LT due to liver mass causing respiratory failure, infiltration of the bladder wall was found. 

Two patients had a positive margin (R1), the first one after urgent LT with infiltration of the bladder wall. The second patient with an R1 margin had malignant thrombosis of a portal vein before transplantation. During DDLT, anastomosis of the portal vein to the confluence of the splenic vein and superior mesenteric vein was performed, and a positive microscopic margin was detected in the wall of a portal vein. Total resection with an R0 margin was achieved in 44 children (95.7%). 

Details of tumor histology are listed in Table 1. Angioinvasion, defined as the presence of tumor cells in the vessels, was identified in 16 patients (34.8%). In the HB group, only one patient with Abernethy syndrome had cirrhosis.

#### 3.1.3. Outcome

The HB group’s median follow-up time was 6.2 years (IQR: 2.3 to 9.5 years). Forty-two patients (91.3%) received post-transplant chemotherapy.

Tumor recurrence was observed in seven patients (15.2%) after LT (Figure 1). 

The time from LT to relapse was 7, 2, 3, 2, 1, 14, and 2 months. Table 3 shows the risk factors for tumor recurrence present in those patients. Three patients were classified as PRETEXT IV, five had multifocal tumors, and lung metastases were detected in three at diagnosis. Three of those patients underwent rescue LT, and one needed urgent LT. In the HB group, all patients after rescue LT and the patient who underwent urgent LT died. 

Angioinvasion was found in six cases. One had a small-cell undifferentiated (SCU) tumor, and one had the macrotrabecular subtype. Also, the patient with SCU had an AFP level of 2.3 ng/mL at diagnosis. The sites of relapse are detailed in Table 3. The most common site of recurrence was the lungs (six patients, 85.7%), followed by bones (three patients, 42.9%) and abdominal cavity (two patients, 28.6%). In five, recurrence was detected in multiple sites. Two patients underwent thoracotomy and resections of metastases from the lungs: the first patient three months after LT and the second 14 months after LT. Four received chemotherapy for recurrence. In one patient who underwent resection of metastases from the lungs 14 months after LT, radiation therapy to the lungs was administered after metastasectomy. Six patients (85.7%) with recurrence after LT died after 26, 7, 13, 4, 2, and 8 months. One (14.3%) is alive 49 months after LT without evidence of disease. 

In patients with HB after LT, complete remission was achieved in 39 patients (84.8%). One patient developed biliary complications and needed a second and third liver transplantation. This patient died 120 months after the first LT from autoimmune thrombocytopenia. Three patients with remission died due to post-transplant lymphoproliferative disease (PTLD), septic complications, and other causes 6, 31, and 207 months after LT, respectively. 

### 3.2. HCC Group (Table 4)

The 26 patients who underwent LT due to HCC were divided into groups according to transplantation from a deceased donor (DDLT 14 patients, 53.8%) or from a living donor (LDLT 12 patients, 46.2%). 

**Table 4 children-11-00193-t004:** Demographics and characteristics of 26 patients with HCC after LT by donor type.

	All *n* = 26	DDLT *n* = 14 (53.8%)	LDLT *n* = 12 (46.2%)	*p*-Value
Age at diagnosis (years)				
Median (IQR)	7.9 (4.8–12.7)	12.6 (7.5–14.2)	6.6 (3.8–7.9)	0.02
Gender	12 female (46.2%)	7 female (50%)	5 female (41.7%)	0.67
Underlying disease	9 (34.6%)	5 (35.7%)	4 (33.3%)	0.899
Tyrosinemia	5	2	3	
Alagille syndrome	1	1	0	
A1AD and hepatitis C virus infection	1	1	0	
Hepatitis B virus infection	1	1	0	
Biliary atresia	1	0	1	
Rescue LT	4 (15.4%)	2 (14.3%)	2 (16.7%)	0.87
Metastases after liver resection and before rescue LT	2 (7.7%)	1 (7.1%)	1 (8.3%)	
AFP at diagnosis (ng/mL)				0.033
Median (IQR)	9700 (171–84,377)	271 (5–21,337)	29,561 (12,400–137,572)	
<100 ng/mL	5	4	1	
100–1000 ng/mL	5	4	1	
AFP pretransplant (ng/mL)				
Median (IQR)	357 (43–2543)	161 (21–1570)	1345 (247–3652)	0.157
Tumor characteristics				
PRETEXT I	4 (15.4%)	1 (7.1%)	3 (25%)	
PRETEXT II	3 (11.5%)	3 (21.5%)	0	
PRETEXT III	10 (38.5%)	5 (35.7%)	5 (41.7%)	
PRETEXT IV	9 (34.6%)	5 (35.7%)	4 (33.3%)	
Multifocal tumor	16 (61.5%)	9 (64.3%)	7 (58.3%)	0.76
Tumor rupture before LT	1 (3.8%)	0	1 (8.3%)	0.27
Chemotherapy before LT	19 (73.1%)	11 (78.6%)	8 (66.7%)	0.49
Time from diagnosis to LT (months)				
Median (IQR)	4.8 (2.4–12)	6.5 (2.4–15)	3.48 (2.3–6)	0.336
Waitlist time (days)				
Median (IQR)	37 (9–49)	42 (20–56)	20 (7–35)	0.157
Age at LT (years)				
Median (IQR)	8.6 (5–14.5)	13.8 (8.3–16.8)	5.9 (3.7–8.4)	0.003
Graft type				
Left lateral segments	8 (30.8%)	0	8 (66.7%)	
Left lobe	4 (15.4%)	0	4 (33.3%)	
Whole liver	14 (53.8%)	14 (100%)	0	
Duct to duct biliary anastomosis	12 (46.2%)	6 (41.4)	5 (41.7%)	0.95
ABO-incompatible LT	1 (3.8%)	0	1 (8.3%)	0.27
R0 margin	26 (100%)	14 (100%)	12 (100%)	
Lymph node metastasis	2 (7.7%)	2 (14.3%)		
Fibrolamellar HCC	4 (15.4%)	3 (21.5%)	1 (8.3%)	0.36
Angioinvasion	12 (46.2%)	8 (57.1%)	4 (33.3%)	0.22
Cirrhosis	8 (30.8%)	4 (28.6%)	4 (33.3%)	0.79
Chemotherapy after LT	19 (73.1%)	9 (64.3%)	10 (83.3%)	0.27

A1AD: alpha-1 antitrypsin deficiency.

#### 3.2.1. Pretransplant Data

The median age at diagnosis was 7.9 years (IQR: 4.8 to 12.7 years). Twelve recipients (46.2%) were female. In nine patients (34.6%), HCC was associated with pre-existing liver disease. The most common underlying disease was tyrosinemia, in five patients. One patient had Alagille syndrome, one biliary atresia, one alpha-1 antitrypsin deficiency with concomitant hepatitis C virus infection, and one hepatitis B virus infection. 

Twenty-two patients (84.6%) underwent primary LT. In four (15.4%), rescue LT was performed (Table 2), in all after previous right extended hemihepatectomy. In two of them, additional resection of metastases was performed after hepatic resection and before LT. One underwent resection of a focal lesion in the caudate lobe. In one case, after pathologic confirmation of a positive margin (R1) after hepatic resection, we immediately proceeded with LDLT (four days after resection).

The median AFP at diagnosis was 9700 ng/mL (IQR: 171 to 84,377 ng/mL). The median AFP before LT was 357 ng/mL (IQR: 43 to 2543) and was significantly higher in patients who underwent transplantation from living donors (*p* = 0.033). Five patients had AFP below 100 ng/mL, and five between 100 and 1000 ng/mL. 

Nine patients (36.4%) were classified as PRETEXT IV, ten (38.5%) were PRETEXT III, three (11.5%) were PRETEXT II, and four (7.1%) were PRETEXT I. Three patients with PRETEXT I had tyrosinemia, and one had biliary atresia. Two patients with PRETEXT II had underlying diseases (A1AT and HCV, and tyrosinemia). The tumor in the third patient with PRETEXT II was localized in segment IV near the liver hilum, and the fibrolamellar variant was confirmed after a percutaneous liver biopsy. A multifocal tumor was found in 16 children (61.5%). No patients had metastases detected at the time of diagnosis. In one, the first sign of liver tumor was abdominal bleeding caused by tumor rupture. The patient underwent emergency laparotomy with partial resection of segment VI. 

Nineteen patients (73.1%) received preoperative chemotherapy. 

#### 3.2.2. Transplant Data and Tumor Pathology

The median time from diagnosis to LT was 4.8 months (IQR: 2.4 to 12 months), and the median waitlist time was 32 days (IQR: 9 to 49 days). No significant differences between the DDLT and LDLT groups were found in time from diagnosis to LT and waitlist time (*p* = 0.34 and *p* = 0.16, respectively). The median age at LT was 8.6 years (IQR 5 to 14.5 years). Patients who underwent LDLT were significantly younger than patients who had a DDLT (5.9 vs. 13.8 years, *p* = 0.003).

Eight patients with HCC received left lateral segment LDLT, and four received left lobe LDLT. All patients undergoing DDLT received a whole liver. ABO-incompatible LT was performed only in one patient who underwent transplantation from a living donor. 

In two patients (7.7%) with the fibrolamellar variant during hepatectomy, metastases to lymph nodules localized in the hepatoduodenal ligament were found. 

Total resection with R0 margin was achieved in all children. Four patients (15.4%) had the fibrolamellar variant. Angioinvasion was present in 12 patients (46.2%). Liver cirrhosis was confirmed in eight of the nine patients with underlying diseases: five with tyrosinemia, one with Alagille syndrome, one with biliary atresia, and one with alpha-1 antitrypsin deficiency with viral hepatitis C. Patients with viral hepatitis B had normal liver without cirrhosis. 

#### 3.2.3. Outcome 

The median follow-up time in the HCC group is 10.4 years (IQR: 3.2 to 19.8 years). Nineteen patients (73.1%) received post-transplant chemotherapy. Six of the seven patients without chemotherapy after LT had underlying diseases. 

Tumor recurrence in patients with HCC after LT was observed in four patients (15.2%) (Figure 2). Three patients had undergone DDLT, and one received LDLT. The time from LT to relapse was 18, 20, and 2 months and was unknown in one patient. Three patients after DDLT with recurrence of HCC died 27, 24, and 46 months after LT. One patient after LDLT is alive with disease. 

Risk factors for tumor recurrence present in HCC patients with relapse are shown in Table 3. One patient was classified as PRETEXT IV and in two, multifocal tumors were detected. Two patients with relapse underwent rescue LT. In three children, the HCC fibrolamellar variant was present. Angioinvasion was found in three cases. The sites of recurrence are listed in Table 3. Two patients who developed recurrence underwent surgical treatment and received chemotherapy combined with sorafenib. One patient underwent two laparotomies and resections of metastases from the abdominal cavity 20 and 34 months after LT. The patient developed bone metastases 39 months after LT and died 7 months later due to tumor progression. In the second patient, pulmonary metastases were detected two months after LDLT, and the patient was treated with pulmonary metastasectomy 2, 2, and 9 months after LT. The patient is alive with the disease. 

In patients with HCC after LT, complete remission was achieved in 22 patients (84.6%). Liver retransplantation was performed in two patients, both due to biliary complications. One patient needed a second transplantation four years after LT. One needed a second transplantation seven years after LT and, four years later, underwent a third liver transplantation due to chronic rejection. Both patients after liver retransplantation are alive. One patient died four days after LT due to multiple organ dysfunction syndrome (MODS), one died 56 months after LT due to post-transplant lymphoproliferative disease (PTLD), and one died five months after LT due to chronic rejection. 

Depending on donor type, estimated patient survival after 5, 10, and 15 years was 63%, 63%, and 63% after DDLT and 100%, 90%, and 90% after LDLT, respectively (Figure 3). In the HCC group undergoing LDLT and DDLT, there was a tendency for better results in the LDLT group, but it did not reach statistical significance (*p* = 0.09). 

### 3.3. Comparison of Outcome between HB and HCC Groups

Estimated patient survival after 5, 10, and 15 years was 82%, 73%, and 73% in the HB group and 79%, 75%, and 75% in the HCC group, respectively (Figure 4). Patient survival was similar in both groups (*p* = 0.75667). A comparison of estimated event-free survival (EFS) revealed no significant differences between the groups (*p* = 0.94). 

### 3.4. Impact of Risk Factors on Patient Survival

In our cohort, 18 patients had no risk factors, 13 patients had one risk factor, 17 patients had two risk factors, 17 patients had three risk factors, 4 patients had four risk factors, and 3 patients had five risk factors. No recipient with five risk factors survived. Estimated patient survival with four risk factors was inferior to patients with one, two, or three risk factors. Estimated recurrence-free survival in patients who had three or fewer risk factors was significantly better than in patients with more than three risk factors (log-rank test *p* = 0.0001) (Figure 5).

In the univariate analysis, extrahepatic disease, angioinvasion, rescue LT, unfavorable histology, and presence of >3 risk factors were significantly correlated with recurrence risk (Table 5). While constructing a multivariable model, we used R^2^ and the Akaike information criterion test as measures of fit of our model. However, we were able to include only two variables due to the collinearity of the remaining factors. Both extrahepatic disease and angioinvasion were considered risk factors for disease recurrence in multivariable analysis.

## 4. Discussion

The key decision in surgical management of patients with advanced HB or HCC is the choice between surgical resection and LT. In the study conducted by Fuchs et al., 5-year survival after extended surgical resection in 27 HB patients with POSTEXT III and IV was 80.7% [12]. However, four patients developed biliary complications, and tumor recurrence was observed in eight patients of whom four died. The study also found that in selected patients, backup LT should be prepared for possible liver failure after resection. We previously demonstrated better outcomes after primary LT in HCC than after surgical resection (72% versus 40%) [13]. Similarly, McAteer et al. reported that 53.4% of patients survived five years after surgical resection compared with 85.3% after LT [14]. 

Our strategy to manage patients with HB or HCC is to avoid resection of the liver tumor when it poses an increased risk of leaving a positive margin or could lead to failure of the liver remnant. Every patient with liver tumor PRETEXT III and IV should be considered a potential candidate for LT. In our study, we demonstrated similar outcomes for LT in unresectable HB and HCC tumors, with good long-term patient survival and event-free survival compared with other studies [2,4,7].

Patients with incomplete surgical resection and recurrence may need rescue LT. The outcome of rescue LT has improved over the years, but it is a treatment option in selected cases only. In the study of Otte et al., the incidence of rescue LT in HB patients was 28%, with overall survival 30% after 6 years [15]. Recent studies have demonstrated better outcomes in patients after rescue LT, e.g., Umeda et al. reported on 24 patients after LT due to HB. Eleven underwent salvage LT, and 5-year overall survival was similar in both groups (72.7% and 69.2%) [9]. Data from the Society of Pediatric Liver Transplantation (SPLIT) showed a cohort of 13 patients with HB after rescue LT with an event-free survival rate of 85% and 74% at 1 and 3 years after transplantation [6]. In our study, all three children after rescue LT for HB recurrence died. In the first patient, LT was performed at the beginning of our transplant program. In the second patient, suspicion of bone metastases (lumbar compression fracture) was present, but we could not confirm it at that time. After LT, we observed recurrence in the lumbar vertebrae and lungs. The third patient, after right trisegmentectomy, underwent two thermoablation procedures for focal lesions in the liver shortly after resection, and, during LT, infiltration of the spleen was found. In a group of 163 patients with HCC, the reported incidence of rescue LT was 9.2% [1]. In our group of 26 HCC patients, the incidence of rescue LT was higher. In both groups, the most common indication for rescue LT was tumor recurrence in the liver after surgical resection, followed by margin-positive resection. Vinayak et al. described a group of 25 patients who underwent LT due to HCC [16]. Four of them received rescue LT and none survived. In our study, two patients are alive in remission and two relapsed after rescue LT. In patients with remission after LT, recurrence was confined to the liver. In two patients with HCC recurrence after rescue LT, we found metastatic disease outside the liver before LT.

Aronson et al. evaluated the impact of a positive microscopic margin after surgical treatment on recurrence and patient survival in a group of 431 patients with HB [17]. They showed a similar 5-year outcome in patients with complete resection and a positive microscopic margin (5-year survival was 92% in patients with complete resection and 91% with a positive margin. The 5-year event-free survival was 86% in both groups). But in their conclusion, the authors underlined that the aim of surgical management was complete resection with a negative microscopic margin. The results of the International Childhood Liver Tumor Strategy Group (SIOPEL 2 and SIOPEL 3) trials demonstrated that in pediatric HCC, only patients after complete surgical resection could survive [3]. In the current study, complete surgical resection with total hepatectomy and LT was obtained in all patients with HCC and 95.7% of cases of HB. One patient with a positive microscopic margin is alive after chemotherapy, and one has died. 

In our study, three patients (6.5%) with HB and seven patients (26.9%) with HCC underwent LT for PRETEXT I/II. In other studies, the reported incidence of LT in patients with HB and PRETEXT I/II was either similar to our group or two times higher (15.1%) [6,7]. Pire et al. described a group of 10 patients with HCC who underwent LT, four of whom presented with PRETEXT II [7]. Eight of these patients had underlying disease, however. In large cohorts of patients with HCC after LT, underlying disease was present in 80.9% of patients [1]. In our study, only 34.6% of patients with HCC had underlying disease. In pediatric patients with HB or HCC and chronic liver disease with progression to liver cirrhosis, LT is the treatment of choice. In our group of patients presenting with PRETEXT I/II, one patient with HB and six patients with HCC had chronic liver disease, and liver resection was not attempted. In all of the patients, cirrhosis was confirmed on pathologic examination. Two patients with HB and PRETEXT I/II were qualified for LT due to tumor recurrence after previous hepatic resections in another hospital. In one patient with HCC and the fibrolamellar variant, the tumor was localized close to the liver hilum. In this patient, partial liver resection was not attempted due to the high risk of R1 resection. Generally, our policy, supported by much better outcomes, is to proceed to liver transplantation in HCC > PRETEXT I rather than liver resection [13].

In our cohort, ten patients died after LT (21.7%) in the HB group and six (23.1%) in the HCC group. Similar to other studies, the most common cause of patient death was tumor recurrence [4]. Recurrent malignancy accounted for 60% of deaths in the HB group and 50% of deaths in the HCC group. Tumor recurrence was observed during the first two years after LT, and lungs were the most common site of recurrence. In our cohort, relapse was observed in two patients with HB and infiltration of other organs, two patients with HCC and lymph node metastases, and two with HCC after surgical resection of metastases located outside of the liver prior to LT. Additionally, three patients with HCC who died due to recurrence had the fibrolamellar variant. In contrast, Kakos et al. showed no difference in patient survival between nonfibrolamellar and fibrolamellar HCC [1]. 

Treatment options for recurrent tumors after LT include chemotherapy and surgical resection, but prognosis and outcome are poor. An earlier study demonstrated that only 14.7% of patients with recurrence of HB after LT survived without disease [9]. In the current study, one of the seven patients with HB relapse after LT is alive and free from disease, and one of the four patients with HCC recurrence is alive with disease. In the patients with tumor recurrence, treatment was individualized. If the disease was localized, surgical resection of metastases was attempted. There was no standard chemotherapy protocol for patients with tumor recurrence. Patients received different numbers of courses of chemotherapy with varying drugs depending on their clinical condition. In selected patients with HCC, sorafenib was used as second-line therapy. One patient with HB and lung metastases received radiation therapy. 

Boster et al. presented a low rate of transplantation from living donors in HB and HCC [6]. Only 3.8% patients of 157 with HB and 16.7% patients of 18 with HCC underwent LDLT. Pire et al. presented a cohort of 45 patients with hepatic tumors after LDLT [7]. Thirty-three of them underwent LDLT due to HB and ten due to HCC. Of these, 87.9% of patients with HB and 80% with HCC achieved remission. In our study, in the HB group, most children (89.1%) underwent LDLT. In the HCC group, 46.2% received a graft from a living donor. The disadvantages of LDLT in liver tumors may be associated with obtaining complete surgical resection if the tumor is located very close to the inferior vena cava or confluence of the hepatic vein. The solution to this problem is total hepatectomy with the inferior vena cava and reconstruction with a venous graft. Pire et al. reported that the inferior vena cava was removed in 81.8% of LTs in patients with HB and in 40% of LTs in patients with HCC [7]. For venous reconstruction, these authors used the left internal jugular vein harvested from a living donor. In our cohort, in only four patients with HB did we decide on vein replacement using frozen iliac vein grafts. Our patients with HCC who underwent LDLT were younger with a higher AFP. In the HCC group, LDLT and DDLT provided similar patient survival. Having a potential living donor expanded the donor pool, especially in younger recipients, decreased waitlist time, and allowed for optimal timing of LT after chemotherapy. LDLT may contribute to better patient survival in children with HB or HCC. 

The impact of individual risk factors for recurrence and patient survival after LT has been widely discussed, and in some studies, it was impossible to prove their significance on patient outcomes [1,6,7,8,9]. In our analysis, we assessed the impact of a number of known risk factors for recurrence. In patients with three or fewer risk factors, survival without recurrence after LT in both HB and HCC patients was excellent. When the number of risk factors increased to more than three, recurrence risk became high, and patient survival was poor. In all patients with malignant liver tumors, qualification for LT and overall and oncological outcome should be discussed within the transplant team and with parents/guardians. We think that in the presence of more than three factors, LT should be considered as having a very high risk of tumor recurrence. 

In patients with HB or HCC and the presence of active extrahepatic disease, LT should be contraindicated. It should be noted, however, that reaching such a diagnosis may be challenging. In our group, in two patients with HB, infiltration of another organ was not seen in radiologic studies and was discovered during hepatectomy. Similarly, two patients with HCC metastases to lymph nodes were found during LT. In some patients prepared for LDLT, we performed exploration before donor surgery with intraoperative use of indocyanine green fluorescent navigation to assess disease extent, and, in some cases, we found intraabdominal metastases (diaphragm, omentum, and lymph nodes). Such a protocol may help to make a final decision on proceeding with transplantation or not. The same protocol may be used in DDLT with exploration of the potential recipient after the offer of the DD donor is accepted. 

## 5. Conclusions

Adequate patient selection is necessary when considering LT for primary liver tumors in children. The presence of more than three risk factors is associated with a very high risk of recurrence and indicates poor prognosis, whereas extrahepatic disease may be considered a contraindication for transplantation. 

## Figures and Tables

**Figure 1 children-11-00193-f001:**
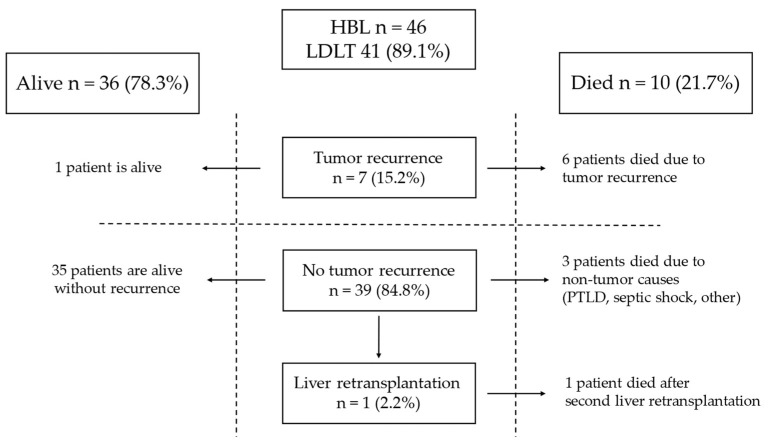
Outcome of patients with HB.

**Figure 2 children-11-00193-f002:**
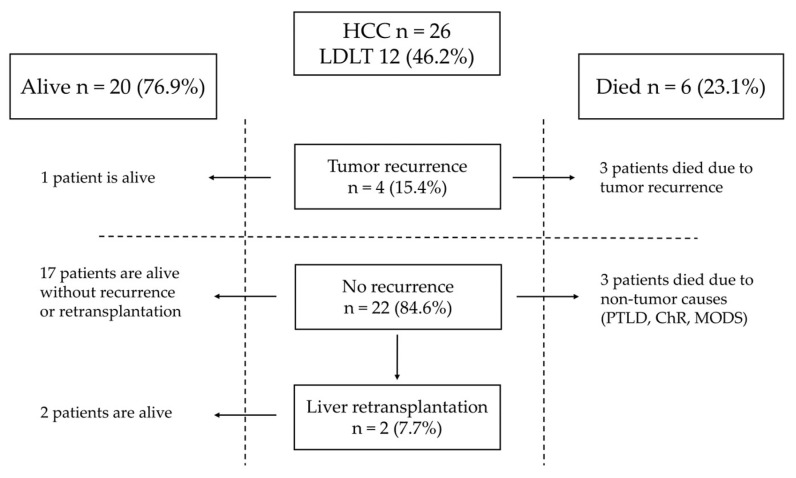
Outcome of patients with HCC.

**Figure 3 children-11-00193-f003:**
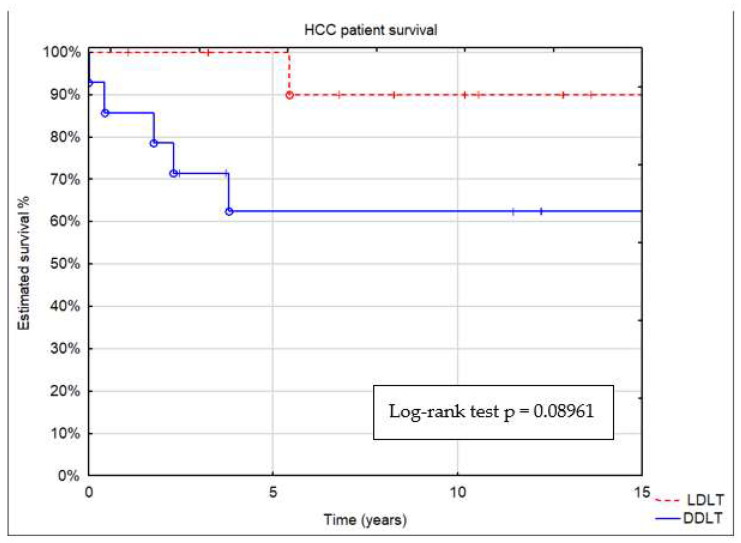
Patient survival by donor type: deceased (DDLT) or living (LDLT).

**Figure 4 children-11-00193-f004:**
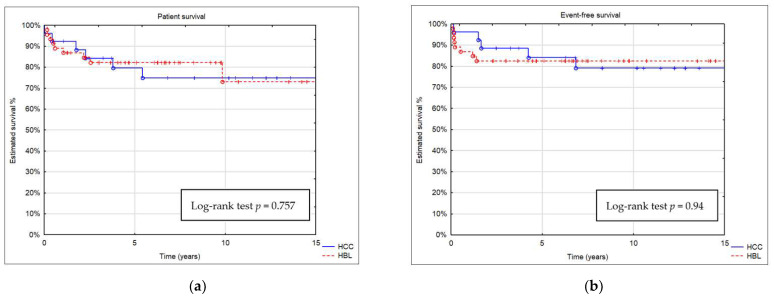
Kaplan–Meier curve showing (**a**) patient survival and (**b**) event-free survival (EFS) of patients with HB and HCC.

**Figure 5 children-11-00193-f005:**
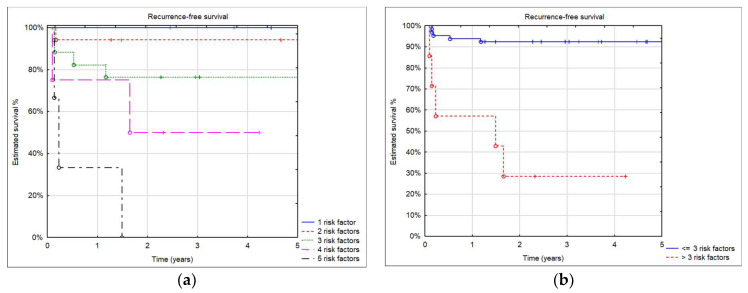
(**a**) 5-year recurrence-free survival by number of risk factors; (**b**) 5-year recurrence-free survival in patients with three and fewer risk factors and more than three.

**Table 2 children-11-00193-t002:** Surgical treatment before rescue liver transplantation and outcome.

	PRETEXT	AFP at Diagnosis	Surgical Treatment before LT	Time before LT (mos.)	Outcome (mos.)
HB					
1	2	600	Extended right hemihepatectomy	24	DOD (26)
2	2	365,000	Extended right hemihepatectomy	10	DOD (7)
3	3	541,854	Extended right hemihepatectomy	5	DOD (4)
			Thermoablation of focal lesions in liver	2	
			Thermoablation of focal lesions in liver	1	
HCC					
4	3	2.3	Extended right hemihepatectomy	96	DOD (27)
			Metastatectomy, hepatoduodenal ligament	48	
			Metastatectomy, paraaortic lymph nodule	24	
			Metastatectomy, hepatoduodenal ligament	6	
5	3	1200	Extended right hemihepatectomy	72	NED (237)
			Resection of focal lesion in caudate lobe	6	
6	3	181,000	Extended right hemihepatectomy	4 days	NED (81)
7	3	137,562	Extended right hemihepatectomy	9	AWD (13)
			Metastasectomy, diaphragm	1 day	

DOD: died of disease, NED: no evidence of disease, AWD: alive with disease.

**Table 3 children-11-00193-t003:** Characteristics and clinical outcome of patients with tumor recurrence.

	Risk Factors for Recurrence	Duration from LT to Relapse (mos.)	Site of Relapse	Surgical Treatment (Number of Procedures)	Chemotherapy for Recurrence after LT	Outcome (mos.)
HB						
1	Tumor multifocality Rescue LT Angioinvasion	7	Bones	None	Yes	DOD (26)
2	Lung metastases at diagnosis Rescue LT	2	Bones Lung	None	Yes	DOD (7)
3	PRETEXT IV Lung metastases at diagnosis Tumor multifocality Anaplastic tumor Angioinvasion	3	Lung	Thoracotomy and metastasectomy (2)	Yes	DOD (13)
4	Lung metastases at diagnosis Tumor multifocality Rescue LT Infiltration of spleen during LT Angioinvasion	2	Bones Lung	None	No	DOD (4)
5	Urgent LT due to respiratory failure caused by tumor mass AFP < 100 ng/mL Infiltration of bladder wall during LT SCUD Angioinvasion	1	Lung Abdominal cavity	None	No	DOD (2)
6	PRETEXT IV Tumor multifocality Angioinvasion	14	Lung	Thoracotomy and metastasectomy (1)	Yes	NED (49)
7	PRETEXT IV Tumor multifocality Angioinvasion	2	Lung Abdominal cavity	None	No	DOD (8)
HCC						
1	Rescue LT Metastasectomy before LT Fibrolamellar HCC	NA	NA	None	No	DOD (2s7)
2	PRETEXT IV Tumor multifocality Fibrolamellar HCC Extrahepatic disease Angioinvasion	18	Lung Mediastinum	None	No	DOD (24)
3	Tumor multifocality Fibrolamellar HCC Extrahepatic disease Angioinvasion	20	Abdominal lymph nodule Bones	Laparotomy and metastasectomy (2)	Yes	DOD (46)
4	Rescue LT Metastasectomy before LT Angioinvasion	2	Lung	Thoracoscopy and metastasectomy (3)	Yes	AWD (13)

DOD: died of disease, NED: no evidence of disease, AWD: alive with disease.

**Table 5 children-11-00193-t005:** Cox proportional hazard model—recurrence risk factors analysis.

Characteristic	Univariate Analysis			Multivariable Analysis	
	HR	X^2^	*p*	HR (95% CI)	*p*
Age at LT	1.1	3.2	0.07		
Pretext IV	0.9	0.02	0.89		
Metastasis	3.0	2.54	0.11		
Multifocal tumor	1.8	0.92	0.33		
Tumor rupture	0	0	0.99		
Extrahepatic disease	12.5	17.1	<0.0001	8.83 (2.65–29.4)	<0.0004
Angioinvasion	9.1	7.82	0.005	7.36 (1.40–38.71)	0.02
Rescue LT	10.4	14.3	<0.0002		
Unfavorable histology	8.6	12.4	<0.0005		
>3 risk factors	13.0	16.2	<0.0001		

## Data Availability

Most of the relevant data are contained in the paper. Most of the output data were taken from the Polish National Transplant Registry at https://rejestrytx.gov.pl/tx/ (accessed on 3 January 2023). Since the data collected in the Registry are sensitive and, thus, protected by law (the Personal Data Protection and the Medical Records Act), access to the database is limited; it can be accessed only upon meeting registration criteria. The Registry is under the supervision of the Polish Transplant Coordinating Centre “Poltransplant”, funded through the budget of the Polish Ministry of Health. All of the patients’ medical histories and other vital information used in the creation of the database and the database itself are available directly at the Children’s Memorial Health Institute, Warsaw, Poland, after contact with the Director of Scientific Affairs, MD PhD Piotr Socha at p.socha@ipczd.pl or corresponding author at p.kalicinski@ipczd.pl.

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
