# Peer review of "Risk for Recurrence in Long-Term Follow-Up of Children after Liver Transplantation for Hepatoblastoma or Hepatocellular Carcinoma"

_children, 2024, doi:10.3390/children11020193_

Round 1
Reviewer 1 Report
Comments and Suggestions for Authors
In this article the authors review their series of liver transplantation in pediatric tumors such as hepatoblastoma and hepatocarcinoma. The main weakness of the article is the lack of novelty in the subject, since there are multicenter series of American national databases with hundreds of patients.
Regarding the structural aspects, the introduction should not start sentences with numbers ( 21% of patients with HB require LT and in less than 20% 34 pediatric patients with HCC resection at diagnosis is possible [2,3]).
Material and methods
The definition of DDLT is missing in material and methods.
What is the reason for periodically performing CT scans in these patients? (CT of the abdomen and thorax was performed 6 91 months after and 1 year after LT or if clinically indicated).
Results
What underlying diseases were present in these patients? (Four patients (8.75%) had an underlying disease. ). It should be noted that these results are in Table 1.
I was struck by the fact that in both the HB and HCC groups, several patients are PRETEXT I and II. What is the indication to transplant these patients? Why was liver resection not attempted in these cases?
Comments on the Quality of English LanguageModerate editing of English language required
Author Response
Response to reviewer comments
Authors: Marek Stefanowicz, Piotr Kaliciński, Hor Ismail, Adam Kowalski, Dorota Broniszczak, Marek Szymczak, Katarzyna Pankowska-Woźniak, Anna Roszkiewicz, Ewa Święszkowska, Diana Kamińska, Sylwia Szymańska and Grzegorz Kowalewski
Article Title: Risk for recurrence in long-term follow-up of children after liver transplantation for hepatoblastoma or hepatocellular carcinoma.
Dear reviewers:
I want to thank you for taking the time to assess our article. We greatly appreciate the thorough and thoughtful comments provided. Your remarks significantly improved our manuscript. We have ensured that each comment was addressed carefully and the paper is revised accordingly.
Attached below are detailed responses to the reviewer’s comments. The latter are
shown in black, and our responses are in red. Please let us know if you still have any questions or concerns about the manuscript. We will be happy to address them promptly.
Sincerely,
The authors
The manuscript underwent English revision by a native speaker.
Material and methods
The definition of DDLT is missing in material and methods.
We provided the definition of DDLT: deceased donor LT (DDLT).
What is the reason for periodically performing CT scans in these patients? (CT of the abdomen and thorax was performed 6 months after and 1 year after LT or if clinically indicated).
It is a standard protocol of our oncologic follow-up with radiologic surveillance for local recurrence and pulmonary metastases after the last course of adjuvant chemotherapy and one year after surgical resection (or total hepatectomy + LT) in patients with HB and HCC. The time for recurrence in patients with immunosuppressive treatment is usually fast. It was less than one year in most patients with HB and some with HCC.
Results
What underlying diseases were present in these patients? (Four patients (8.75%) had an underlying disease. ). It should be noted that these results are in Table 1.
We listed underlying diseases in the text and Table 1.
I was struck by the fact that in both the HB and HCC groups, several patients are PRETEXT I and II. What is the indication to transplant these patients? Why was liver resection not attempted in these cases?
In patients from HB group two (4.3%) were classified as PRETEXT II, and one (2.2%) was PRETEXT I. Patient with PRETEXT I and Abernethy syndrome, underwent embolization of portosystemic shunt three years before LT and later developed cirrhosis and HB tumor. In this patient, liver resection was not attempted due to liver cirrhosis. Two patients with PRETEXT II were qualified for LT due to tumor recurrence after previous hepatic resections in another hospital.
In the HCC group, three patients (11.5%) were classified as PRETEXT II, and four (7.1%) were PRETEXT I. All patients with PRETEXT I had cirrhosis, in three due to tyrosinemia and in one due to biliary atresia. Two patients with PRETEXT II had cirrhosis due to underlying diseases (A1AT and HCV, tyrosinemia). The third patient with PRETEXT II had a 6 cm tumor localized in segment four near the liver hilum. The fibrolamellar variant was confirmed after a percutaneous liver biopsy. The risk of R1 resection was very high in this patient, so he was qualified for LT. Generally, our policy, proved by much better results (published by our team and confirmed by experience from other centers[1,2]), is to proceed to liver transplantation in HCC > PRETEXT I rather than liver resection.
- Ismail, H.; Broniszczak, D.; Kalicinski, P.; Markiewicz-Kijewska, M.; Teisseyre, J.; Stefanowicz, M.; Szymczak, M.; Dembowska-Baginska, B.; Kluge, P.; Perek, D.; et al. Liver transplantation in children with hepatocellular carcinoma. Do Milan criteria apply to pediatric patients? Pediatr Transplant 2009, 13, 682-692, doi:10.1111/j.1399-3046.2009.01062.x.
- McAteer, J.P.; Goldin, A.B.; Healey, P.J.; Gow, K.W. Surgical treatment of primary liver tumors in children: outcomes analysis of resection and transplantation in the SEER database. Pediatr Transplant 2013, 17, 744-750, doi:10.1111/petr.12144.
Reviewer 2 Report
Comments and Suggestions for Authors
This paper is a description of a single center's 30+ years experience with liver transplantation in pediatric malignant liver tumors. The focus is on the surgical aspects of treatment.
General Comments:
1. There is no list of abbreviations. Some abbreviations are not spelled out before using in the text such as AFP, US, CT.
2. Why no multivariate analysis, this would have allowed weighting of the risk factors?
3. Some details of the chemotherapy regimens employed, especially post-transplant chemotherapy, would be helpful in understanding outcomes. For example, were they relatively standard and was there any differences between those that survived and those that had recurrence?
4. The comparison between DDLT and LDLT for the HCC group is concerning regarding the statement made in the abstract and the discussion that LDLT provided better survival. Not only do the survival curves not show significance, there is no consideration for the difference between the cohorts in terms of PRETEXT, age and underlying disease. Again, a multivariate analysis would help here.
5. How is this analysis going to alter the authors approach to future cases? Will they be even more selective in attempting rescue LT? Will they consider LT futile for those with 4 or 5 risk factors and if not, what will they do differently?
Specific Comments:
Abstract:
1. Line 22: Does a non-significant p-value need to be quoted to 5 decimal places?
Introduction:
2. Line 32: HB is the most common primary malignant liver tumor. Several benign liver tumors are more common.
3. Lines 34-35: “and in less than 20% of pediatric patients with HCC resection at diagnosis is possible”
Materials and Methods:
4. Line 88: AB0 (ABzero) should be ABO. Also in Table 1, line 143, Table 4, and line 233.
5. Line 89: “Il-2 antagonist” does this mean an IL-2 receptor antagonist such as basiliximab?
Results:
6. Line 179: “the first patient three and months after LT” should this be “three months”?
Discussion:
7. Line 378: “Boster” not “Bolster”
Author Response
Response to reviewer comments
Authors: Marek Stefanowicz, Piotr Kaliciński, Hor Ismail, Adam Kowalski, Dorota Broniszczak, Marek Szymczak, Katarzyna Pankowska-Woźniak, Anna Roszkiewicz, Ewa Święszkowska, Diana Kamińska, Sylwia Szymańska and Grzegorz Kowalewski
Article Title: Risk for recurrence in long-term follow-up of children after liver transplantation for hepatoblastoma or hepatocellular carcinoma.
Dear reviewers:
I would like to thank you for taking the time to assess our article. We greatly appreciate the thorough and thoughtful comments provided. Your remarks significantly improved our manuscript. We have ensured that each comment was addressed carefully and the paper is revised accordingly.
Attached below are detailed responses to the reviewer’s comments. The latter are
shown in black and our responses in red. Please let us know if you still have any questions or concerns about the manuscript. We will be happy to address them in a timely manner.
Sincerely,
The authors
The manuscript underwent English revision by a native speaker.
- There is no list of abbreviations. Some abbreviations are not spelled out before using in the text such as AFP, US, CT.
We spelled out abbreviations used in text.
- Why no multivariate analysis, this would have allowed weighting of the risk factors?
We created a Cox proportional hazard model to analyze known and potential risk factors.
- Some details of the chemotherapy regimens employed, especially post-transplant chemotherapy, would be helpful in understanding outcomes. For example, were they relatively standard and was there any differences between those that survived and those that had recurrence?
After LT, patients with HB and HCC received two or three courses of cisplatin and doxorubicin as a standard treatment.
In patients with tumor recurrence it was very difficult to compare treatment of recurrence. If disease was localized surgical resection of metastases was attempted. There was no standard protocol of chemotherapy for patients with tumor recurrence. Treatment was individualized, and the patient received a different number of courses of chemotherapy with varying drugs depending on clinical condition. In two patients with HCC, sorafenib was used as second-line therapy. One patient with HB after resection of lung metastases received radiation therapy.
- The comparison between DDLT and LDLT for the HCC group is concerning regarding the statement made in the abstract and the discussion that LDLT provided better survival. Not only do the survival curves not show significance, there is no consideration for the difference between the cohorts in terms of PRETEXT, age and underlying disease. Again, a multivariate analysis would help here.
We modified our statement to describe the data adequately. You are right that the difference did not reach statistical significance (p = 0.09). Both groups differed only in age and AFP levels. There were no statistical differences in the PRETEXT and underlying diseases. (lines 462-467)
- 5. How is this analysis going to alter the authors approach to future cases? Will they be even more selective in attempting rescue LT? Will they consider LT futile for those with 4 or 5 risk factors and if not, what will they do differently?
In all patients with malignant liver tumors, qualification for LT and overall oncological outcome should be discussed within the transplant team and with parents/guardians. We think that in the presence of more than three factors, LT should be considered as having a very high risk of recurrence.
LT should be contraindicated in patients with HB or HCC and the presence of active extrahepatic disease. It should be noted, however, that reaching such a diagnosis may be challenging. In our group of two patients with HB, infiltration of another organ was not seen in radiologic studies and was discovered during hepatectomy. Similarly, in two patients with HCC, metastases to lymph nodes were found during LT. We have done this in some patients prepared for LDLT exploration before donor surgery with intraoperative use of indocyanine green fluorescent navigation to assess the extent of disease. In some cases, we’ve found intraabdominal metastases (diaphragm, omentum, lymph nodes). Such a protocol may help decide whether to proceed with transplantation. The same protocol may be used in DDLT with exploration of the potential recipient after the offer of the DD donor is accepted.
Specific Comments:
Abstract:
- Line 22: Does a non-significant p-value need to be quoted to 5 decimal places?
We addressed this issue.
Introduction:
- Line 32: HB is the most common primary malignant liver tumor. Several benign liver tumors are more common.
In the pediatric population, hepatoblastoma (HB) is the most common primary malignant liver tumor, followed by hepatocellular carcinoma (HCC).
- 3. Lines 34-35: “and in less than 20% of pediatric patients with HCC resection at diagnosis is possible”.
A fifth of all patients with HB require LT, and in most pediatric patients with HCC, resection at diagnosis is not possible.
Materials and Methods:
- Line 88: AB0 (ABzero) should be ABO. Also in Table 1, line 143, Table 4, and line 233.
We changed AB0 on ABO.
- Line 89: “Il-2 antagonist” does this mean an IL-2 receptor antagonist such as basiliximab?
Children older than two years with ABO incompatible LT received induction therapy with Il-2 receptor antagonist.
Results:
- Line 179: “the first patient three and months after LT” should this be “three months”?
Two patients underwent thoracotomy and resections of metastases from the lungs: the first patient three months after LT, and the second 14 months after LT.
Discussion:
- Line 378: “Boster” not “Bolster”
We corrected the author's name.
Reviewer 3 Report
Comments and Suggestions for Authors
Authors describe long-term results of LT in pediatric patients hepatoblastoma and hepatocarcinoma, focusing especially on the causes and patterns of tumor recurrence. I have no substantive comments regarding the manuscript, except question if any other types of treatment except chemotherapy were used in patients with relapse - any biological? radiotherapy of lungs?. The manuscript contains all the relevant information. I suggest improving the graphic part. The figure and graphs are blurred and should be corrected. Please consider including graphic abstract.
Author Response
Response to reviewer comments
Authors: Marek Stefanowicz, Piotr Kaliciński, Hor Ismail, Adam Kowalski, Dorota Broniszczak, Marek Szymczak, Katarzyna Pankowska-Woźniak, Anna Roszkiewicz, Ewa Święszkowska, Diana Kamińska, Sylwia Szymańska and Grzegorz Kowalewski
Article Title: Risk for recurrence in long-term follow-up of children after liver transplantation for hepatoblastoma or hepatocellular carcinoma.
Dear reviewers:
I would like to thank you for taking the time to assess our article. We greatly appreciate the thorough and thoughtful comments provided. Your remarks significantly improved our manuscript. We have ensured that each comment was addressed carefully and the paper is revised accordingly.
Attached below are detailed responses to your comments. The latter are shown in black, and our responses are in red. Please let us know if you still have any questions or concerns about the manuscript. We will be happy to address them promptly.
Sincerely,
The authors
The manuscript underwent English revision by a native speaker.
Authors describe long-term results of LT in pediatric patients hepatoblastoma and hepatocarcinoma, focusing especially on the causes and patterns of tumor recurrence. I have no substantive comments regarding the manuscript, except question if any other types of treatment except chemotherapy were used in patients with relapse - any biological? radiotherapy of lungs?. The manuscript contains all the relevant information.
In one patient with HB who underwent resection of metastases from the lungs 14 months after LT radiation therapy of the lungs was administered after metastasectomy. Two patients with HCC who developed recurrence after LT underwent surgical treatment, and received chemotherapy combined with sorafenib.
I suggest improving the graphic part. The figure and graphs are blurred and should be corrected. Please consider including graphic abstract.
We will gladly provide the editorial team with high-quality 300 dpi graphs and include a graphic abstract upon your suggestion.
Round 2
Reviewer 1 Report
Comments and Suggestions for Authors
The authors have responded to the issues raised by the reviewers.
Comments on the Quality of English LanguageThe authors have responded to the issues raised by the reviewers.
Author Response
Thank you for your substantive guidance.
Reviewer 2 Report
Comments and Suggestions for Authors
Thanks to the authors for the revisions.
Two minor questions:
1. In Materials and Methods, line 79: what is "2ur2 analysis"
2. In Discussion, line 465: what is "16ur"
Author Response
We have ensured that all misspellings are eliminated from our manuscript. Thank you for your substantial guidance.